# Knowledge, uptake and intention to use antibiotic post-exposure prophylaxis and meningococcal B vaccine (4CMenB) for gonorrhoea among a large, online community sample of men and gender-diverse individuals who have sex with men in the UK

Dana Ogaz[1,2]*, Jessica Edney[1], Dawn Phillips[1], Dolores Mullen[1], David Reid[2,3], Ruth Wilkie[1], Erna Buitendam[1], James Bell[1,2], Catherine M. Lowndes[1], Gwenda Hughes[4], Helen Fifer[1], Catherine H. Mercer[2,3], John Saunders[1,2], Hamish Mohammed[1,2]

1 Blood Safety, Hepatitis, STI & HIV Division, UK Health Security Agency, London, United Kingdom, 2 The National Institute for Health Research Health Protection Research Unit in Blood Borne and Sexually Transmitted Infections at University College London in Partnership with the UK Health Security Agency, London, United Kingdom, 3 Institute for Global Health, University College London, London, United Kingdom, 4 UK Public Health Rapid Support Team, London School of Hygiene & Tropical Medicine, London, United Kingdom

* dana.ogaz@ukhsa.gov.uk

## Abstract

Novel STI prevention interventions, including doxycycline post-exposure prophylaxis (doxyPEP) and meningococcal B vaccination (4CMenB) against gonorrhoea, have been increasingly examined as tools to aid STI control. There is evidence of the efficacy of doxyPEP in preventing bacterial STIs; however, limited data exist on the extent of use in the UK. We examined self-reported knowledge and use of antibiotic post-exposure prophylaxis (PEP), and intention to use (ITU) doxyPEP and 4CMenB among a large, community sample of men and gender-diverse individuals who have sex with men in the UK. Using data collected by the RiiSH survey (November/December 2023), part of a series of online surveys of men and other gender-diverse individuals in the UK, we describe (%, [95% CI]) self-reported knowledge and use of antibiotic PEP (including doxyPEP) and doxyPEP and 4CMenB ITU. Using bivariate and multivariable logistic regression, we examined correlates of ever using antibiotic PEP, doxyPEP ITU, and 4CMenB ITU, respectively, adjusting for sociodemographic characteristics and a composite marker of sexual risk defined as reporting (in the last three months): ≥5 condomless anal sex partners, bacterial STI diagnosis, chemsex, and/or meeting partners at sex-on-premises venues, sex parties, or cruising locations. Of 1,106 participants (median age: 44 years [IQR: 34–54]), 34% (30%-37%) knew of antibiotic PEP; 8% (6%-10%) ever reported antibiotic PEP use. Among those who did, most reported use in the last year (84%, 73/87) and exclusively used doxycycline (69%, 60/87). Over half of participants reported doxyPEP ITU (51% [95% CI: 47%-56%], 568/1,106) while over two-thirds

**Data Availability Statement:** The data that support the findings of this study have been assessed by the UK Health Security Agency's Office for Data Acquisition and Release as having sensitive personal information and are therefore not publicly available to protect participant privacy. However, some aggregate data may be available upon reasonable request from the UKHSA. Requests can be directed to DataAccess@ukhsa.gov.uk (UKHSA REGG Ref 524).

**Funding:** DR and CHM received funding support as part of The National Institute for Health Research Health Protection Research Unit in Blood Borne and Sexually Transmitted Infections at University College London in partnership with the UK Health Security Agency (https://bbsti.hpru.nihr.ac.uk). The funders had no role in study design, data collection and analysis, decision to publish, or preparation of the manuscript. All other authors received no specific funding for this work.

**Competing interests:** The authors have declared that no competing interests exist.

(64% [95% CI: 60%-69%], 713/1,106) reported 4CMenB ITU. Participants with markers of sexual risk and with uptake of other preventative interventions were more likely to report ever using antibiotic PEP as well as doxyPEP and 4CMenB ITU, respectively. HIV-PrEP users and people living with HIV (PLWHIV) were more likely to report antibiotic PEP use and doxyPEP and 4CMenB vaccination ITU than HIV-negative participants not reporting recent HIV-PrEP use. Findings demonstrate considerable interest in the use of novel STI prevention interventions, more so for 4CMenB vaccination relative to doxyPEP. Fewer than one in ten participants had reported ever using antibiotic PEP, with most using appropriate, evidence-based antibiotics. The use of antibiotic PEP and the report of doxyPEP ITU and 4CMenB ITU was more common among those at greater risk of STIs.

## Introduction

There have been continued increases in bacterial sexually transmitted infections (STIs) (e.g., chlamydia, gonorrhoea, syphilis) among gay, bisexual, and other men who have sex with men (GBMSM) in the UK since the early 2000s [1, 2]. While there has been sustained progress towards HIV elimination through combination prevention interventions, including HIV-PrEP, HIV testing, and Treatment as Prevention (TasP) [3], controlling transmission of other STIs remains challenging. Novel STI prevention interventions have been increasingly considered to aid STI control in key populations, but advocacy for the use of biomedical interventions, including antibiotic prophylaxis and meningococcal B vaccination (4CMenB) for GBMSM, has been mixed. Concerns with widespread implementation include the potential effects on selection of antimicrobial resistant bacteria as well as individual sexual risk perception and resultant behavioural risk compensation [4–6]. In the UK, elimination of the 2022 mpox clade IIb outbreak was jointly achieved with co-produced community messaging and individual-level behavioural change, alongside a targeted vaccination programme. This underlined the importance of a community supported response [7] and rapid deployment of interventions as part of a cadre of combination prevention tools [8]. However, similar approaches for bacterial STI control have been less effective [9], which highlights a need to consider new and novel prevention interventions as part of modern control measures.

The use of self-sourced antibiotics as pre- and post-exposure prophylaxis for STI prevention has been reported in UK GBMSM from as early as 2019 [10, 11], with uptake estimates of 3.6% across a community sample of GBMSM in 2020/2021 [12]. Recent studies have shown a reduction in bacterial STI incidence in GBMSM using doxycycline post-exposure prophylaxis (doxyPEP) [13–17], and a guideline for doxyPEP use in the UK is under development.

Observational studies of 4CMenB vaccination for *Neisseria meningitidis* serogroup B bacteria have shown cross-protection against *Neisseria gonorrhoeae* (gonorrhoea) [18]. Modelling suggests vaccination for GBMSM at risk can be a cost-effective strategy with significant impact on gonorrhoea incidence, while also conferring community-level benefits that could curb incidence, even within conservative efficacy estimates [19]. Real-world effectiveness of 4CMenB has shown a reduction in gonorrhoea incidence of between 22%-47% [14, 20–23]. More recently, recommendations of a gonorrhoea vaccination programme using 4CMenB vaccine, primarily (not exclusively) targeting GBMSM, have been made by the UK's Joint Committee on Vaccination and Immunisation (JCVI) [24].

At the time of writing this article, doxyPEP and 4CMenB vaccination are not yet recommended for STI prevention in the UK. However, in preparation for potential availability of

these novel STI interventions across SHS, we explored data from the most recent round of a serial, cross-sectional survey of men and gender-diverse individuals having sex with men in the UK to examine self-reported knowledge, uptake, and regimens used as antibiotic post-exposure prophylaxis (PEP) for STI prevention. To gauge community interest in these novel STI preventative interventions, we also explored the intention to use doxyPEP and 4CMenB vaccine assuming efficacy and availability in the UK.

## Methods

### Study population and data collection

We conducted a cross-sectional study and analysis of the most recent round (November/December 2023) of the 'Reducing inequalities in Sexual Health' (RiiSH) surveys, a series of cross-sectional surveys examining the sexual health and well-being of a community sample of men and gender-diverse individuals having sex with men in the UK. Methods for RiiSH 2023 were adapted from previous study rounds [25, 26]. Stakeholder engagement with UK community groups prior to implementation was undertaken to review core questions and inform question additions for the 2023 round.

Participants were recruited from 7th November-6th December 2023 through social networking sites (Facebook, Instagram, X) and a geospatial dating application (Grindr). Advertisements hosted by participating recruitment sites directed users to the online survey. Users who met inclusion criteria were asked to take part. An additional survey link was created for dissemination by voluntary and community networks which was cascaded via social networking sites (hereafter, community-cascaded link). Due to survey hosting limitations, we were unable to systematically capture the number of individuals who accessed this survey without participating.

Participants eligible to take part and included in analyses were self-identifying men (cisgender/transgender), transgender women, or gender-diverse individuals assigned male at birth (AMAB), aged ≥16 years, resident in the UK and reporting sex with a man in the last year (hereafter, men and gender-diverse individuals who have sex with men). Gender identity and sex at birth was derived from responses from a two-step question (S1 Appendix). We obtained online consent and there was no incentive to participate.

Data was collected using the Snap Surveys platform (www.snapsurveys.com). Data management and analysis was conducted using Stata 17.0 (StataCorp LLC). Ethical approval for this study was granted by the UKHSA Research and Ethics Governance Group (REGG; ref: R&D 524) and all methods were performed in accordance with guidelines and regulations set by that group. Young people aged 16–17 years were eligible to participant in this survey as this group represent a key population at risk of STI acquisition in the UK and are important to include this research. Parental consent of participants aged 18 or younger was not sought. Ethical guidelines produced by the British Psychological Society and General Medical Council suggest consent from parents should be sought for those under age 16 and those aged 16 and over may be presumed to be able to reach informed consent if the information on the study, and the way that data is collected, stored, and used is clear. This information was included for all potential participants at the beginning of the survey.

### Knowledge and use of antibiotic post-exposure prophylaxis (PEP) for STI prevention

We calculated the percentage (%) and 95% confidence intervals (95% CI) of participants having ever heard of using antibiotic PEP ('Before taking this survey, had you heard about using

antibiotics immediately after sex to prevent STIs other than HIV [e.g., doxy PEP]?') and those that reported use ('Have you ever used antibiotics in this way?').

Among those reporting the use of antibiotic PEP, we examined recency of use ('When did you last use antibiotics in this way?') and antibiotics ever used. In those who had never used antibiotic PEP, we calculated the percentage (%) and 95% CI of those who had considered use ('Have you ever considered taking antibiotics in this way?') (see S1 Appendix for questions).

## Correlates of ever using antibiotic post-exposure prophylaxis (PEP) for STI prevention

We examined bivariate associations to ever using antibiotic PEP. These included sociodemographic characteristics (age-group, gender identity and sex a birth, sexual orientation, ethnic group, country of birth, employment status, educational qualifications, household composition, financial situation), clinical and behavioural characteristics (HIV status and PrEP use history [in last three months], chemsex in the last year, bacterial STI diagnosis in the last three months, SHS visit in the last year, mpox vaccination history), sexual partnerships since August 2023 (e.g. new partnerships, number of partners, meeting location), personal well-being and sexual satisfaction measures. We used mental health and personal well-being indicators derived from the UK ONS that were dichotomised, as per ONS harmonisation standards [27], for measures of low life satisfaction, low life worthwhileness, low happiness, and high anxiety. Agreement (agree/strongly agree) with the statement, 'I feel satisfied with my sex life' was used as a measure of sexual satisfaction in line with questions in Great Britain's National Surveys of Sexual Attitudes and Lifestyles (Natsal, a national probability sample of sexual behaviour) [28].

We also considered a composite marker of sexual risk as a covariable in analyses. This marker was defined as reporting (in the last three months): $\geq 5$ condomless anal sex partners, bacterial STI diagnosis, chemsex, and/or meeting partners at sex-on-premises venues, sex parties, or cruising locations. We used this marker in multivariable analyses to minimise collinearity in adjusted models. This composite was structured as a binary indicator (i.e., yes/no markers of sexual risk in the last three months) and was comprised of individual markers of sexual risk examined in bivariate analyses and previously described as predictors of STI prophylaxis use [10, 12].

Due to small numbers, we grouped categories for sexual orientation (bisexual, straight, or another way), ethnicity (Black, Asian, Mixed, and other), gender minority groups (transgender and nonbinary). Evidence of association was considered where $p < 0.05$. Bivariate (unadjusted) odds ratios (uORs), 95% CIs, and associated p-values derived from the likelihood ratio test (LRT) were calculated.

We next performed sequential, multivariable modelling to examine adjusted associations to ever using antibiotic PEP. Our initial multivariable model included inclusion of all sociodemographic characteristics with bivariate association to ever using antibiotic PEP and based on *a priori* knowledge, our composite marker of sexual risk to minimise collinearity given correlation among individual markers of sexual risk. Adjustments for these sociodemographic characteristics and our composite measure of sexual risk were carried forward to subsequent models assessing associations to select clinical and behavioural characteristics (excluding individual markers of sexual risk included in [or as a subgroup of] our composite measure), personal well-being and sexual satisfaction measures with evidence of bivariate association to ever using antibiotic PEP. Age-group was selected *a priori* for inclusion in all multivariable models. While ethnicity was also considered for inclusion, adjustment was not carried forward where $p > 0.05$ to limit model sparsity given the low number of observations in subgroups outside of White ethnicities. Adjusted odds ratios (aORs), 95% CIs, and (LRT) p-values are presented.

### Correlates of self-reported intention to use doxycycline post-exposure prophylaxis (doxyPEP) and 4CMenB

All participants were asked about their likelihood ('Very unlikely', 'Somewhat unlikely', 'Somewhat likely', 'Very likely', 'I don't know') of doxyPEP uptake, if available and considered safe and effective, and about their likelihood of 4CMenB uptake (referred to as Bexsero in survey questions), if available and with 30–50% effectiveness against gonorrhoea (see S1 Appendix for questions).

We initially aimed to use ordinal logistic regression to assess correlates of ordinal uptake outcomes; however, given heavy skew to the highest positive responses for each intervention, outcome measures were dichotomised ('Very likely' vs all else), where intention to use (ITU) was defined as those 'very likely' to consider intervention uptake. As per methodology above, we examined correlates of doxyPEP and 4CMenB ITU, respectively, using bivariate and sequential, multivariable logistic regression.

## Results

There were 1,322 participants who completed the RiiSH 2023 survey, of whom, 1,106 met participation criteria and were included in analyses (S2 Appendix). Half of all participants were recruited from Grindr (50%), followed by Instagram (23%), Facebook (19%), and community-cascaded links (7%).

Participants primarily resided in England (85% 941/1,106), were of White ethnicity (89% 984/1,106), cisgender male (95% 1,051/1,106) and were UK-born (78% 860/1,106). The median age of participants was 44 years (interquartile range: 34–54), with two-thirds reporting degree-level education (62% 691/1,106). Over three-quarters of participants were employed (78% 868/1,106), while less than half (41% 454/1,106) reported being financially comfortable.

Since August 2023 (i.e., 3-month lookback period), one in five participants had $\geq 5$ male condomless anal sex (CAS) partners (21% 231/1,106) and 8.7% (96/1,106) reported at least one positive bacterial STI test. Additional characteristics are described in Table 1.

### Knowledge and use of antibiotic post-exposure prophylaxis (PEP) for STI prevention

Over a third (34% [95% CI: 30%-37%], 373/1,106) of all participants had ever heard about using antibiotics after sex for STI prevention; 8% (95% CI: 6%-10%, 87/1,106) reported ever having used antibiotic PEP (Fig 1). Among the latter, 84% (73/87) had done so in the last year, where most reported ever using doxycycline (80%, 68/87) and 69% (60/87) specified its exclusive use. One in ten participants were uncertain of which antibiotics they had used previously (11% 10/87), and few reported exclusive azithromycin (2% 2/87) or amoxicillin use (8% 8/87) (S3 Appendix). In the two-thirds (65% 717/1,106) of participants who had never used antibiotic PEP, nearly one in five (18% [95% CI: 16%-21%], 186/1,019) had ever considered use.

### Correlates of ever using antibiotic post-exposure prophylaxis (PEP) for STI prevention

In bivariate analyses, sociodemographic characteristics associated with the use of antibiotic PEP included country of birth (uOR: 1.74 [95% CI: 1.09–2.80]) born outside the UK vs in the UK) and educational qualifications (uOR: 0.50 [95% CI: 0.30–0.84] below degree-level vs degree-level) and were carried forward to multivariable analyses. Those with our composite marker of sexual risk were more likely to report ever using antibiotic PEP (uOR: 3.08 [95% CI: 1.97–4.80]) yes vs no). We also found positive bivariate associations among those reporting

**Table 1. RiiSH 2023 participant characteristics (n = 1,106).**

|  | % (No.) |
|---|---|
| **All RiiSH 2023 participants** | 100% (1,106) |
| **Recruitment site** |  |
| Grindr | 50% (553) |
| Instagram | 23% (253) |
| Facebook | 19% (213) |
| Twitter | 1% (8) |
| Community-cascaded link | 7% (79) |
| **Sociodemographic characteristics** |  |
| Median age at survey completion (interquartile range) | 44 (34–54) |
| Mean age at survey completion (standard deviation) | 44.1 (12.7) |
| **Age-group (3 categories)** |  |
| 16–29 | 14% (151) |
| 30–44 | 38% (416) |
| 45 and over | 49% (539) |
| **Gender identity and sex at birth** |  |
| All other gender identity groups | 5% (55) |
| Cisgender male | 95% (1,051) |
| **Sexual orientation** |  |
| Gay/homosexual | 82% (910) |
| Bisexual, straight, or another way‡ | 18% (196) |
| **Ethnic group** |  |
| White | 89% (984) |
| Black | 2% (17) |
| Asian | 5% (56) |
| Mixed or other | 4% (49) |
| **Country of birth** |  |
| Outside of the UK | 22% (246) |
| UK | 78% (860) |
| **Nation of residence** |  |
| Scotland, Wales, or N Ireland | 15% (165) |
| England | 85% (941) |
| **Employment** |  |
| Not employed | 22% (238) |
| Current employment (full-time, part-time, self-employed) | 78% (868) |
| **Educational qualifications** |  |
| Below degree-level | 38% (415) |
| Degree-level or higher | 62% (691) |
| **Lives with partner(s)** |  |
| No | 63% (693) |
| Yes | 37% (413) |
| **Comfortable financial situation** |  |
| No | 59% (652) |
| Yes (top two quartiles)§ | 41% (454) |
| **Clinical and behavioural characteristics** |  |
| **HIV status and HIV-PrEP use history** |  |
| HIV negative/unknown—Never used HIV-PrEP | 39% (433) |
| HIV negative/unknown—No HIV-PrEP use since August 2023 | 9% (105) |

(*Continued*)

**Table 1.** (Continued)

| | % (No.) |
|---|---|
| HIV negative/unknown—HIV-PrEP use since August 2023 | 38% (425) |
| PLWHIV | 13% (143) |
| **Recreational drug use associated with chemsex in the last year** | |
| No | 93% (1,030) |
| Yes‡ | 7% (76) |
| **Bacterial STI diagnosis since August 2023** | |
| No | 91% (1,010) |
| Yes | 9% (96) |
| **Tried to get an STI test since August 2023** | |
| No | 48% (532) |
| Yes, able to get a STI test | 44% (489) |
| Yes, unable to get a STI test | 8% (85) |
| **SHS visit since December 2022 (last year)** | |
| No | 42% (470) |
| Yes | 58% (636) |
| **Mpox vaccination history (≥1 dose)** | |
| No | 58% (636) |
| Yes | 42% (470) |
| **Partnerships since August 2023** | |
| **Any new male physical sex partner(s)** | |
| No new partners | 32% (351) |
| 1 or more new partners | 68% (755) |
| **Met male physical sex partner(s) at SOP venue, sex party, or cruising location** | |
| No | 85% (945) |
| Yes | 15% (161) |
| **Number of male physical sex partner(s)** | |
| No sex or only virtual sex | 12% (138) |
| 1 partner | 20% (226) |
| 2–4 partners | 25% (277) |
| 5–9 partners | 18% (195) |
| 10 or more partners | 24% (270) |
| **Number of male condomless anal sex (CAS) partner(s)** | |
| No known CAS partners | 35% (383) |
| 1 CAS partner | 24% (270) |
| 2–4 CAS partners | 20% (222) |
| 5 or more CAS partners | 21% (231) |
| **Markers of sexual risk in the last 3 months (composite)¶** | |
| No | 67% (745) |
| Yes | 33% (361) |
| **Personal well-being measures** | |
| **Low life satisfaction** | |
| No | 78% (862) |
| Yes | 22% (242) |
| Not specified | <1% (2) |
| **Low life worthwhileness** | |
| No | 80% (887) |
| Yes | 20% (217) |

(*Continued*)

**Table 1.** (Continued)

| | % (No.) |
|---|---|
| Not specified | <1% (2) |
| **Low happiness (yesterday)** | |
| No | 77% (853) |
| Yes | 23% (251) |
| Not specified | <1% (2) |
| **High anxiety (yesterday)** | |
| No | 63% (695) |
| Yes | 37% (409) |
| Not specified | <1% (2) |
| **Sexual satisfaction** | |
| "I feel satisfied with my sex life"†† | |
| Disagree/do not agree nor disagree | 58% (646) |
| Agree/strongly agree | 42% (460) |
| **Intention to use doxyPEP*** | |
| Very unlikely | 10% (112) |
| Somewhat unlikely | 10% (110) |
| Somewhat likely | 23% (251) |
| Very likely | 51% (568) |
| Don't know | 6% (65) |
| **Intention to use 4CMenB*** | |
| Very unlikely | 7% (78) |
| Somewhat unlikely | 6% (63) |
| Somewhat likely | 19% (210) |
| Very likely | 64% (713) |
| Don't know | 4% (42) |
| **Ever used antibiotic PEP*** | |
| No | 1,109 (92%) |
| Yes‡‡ | 87 (8%) |

‡Includes those identifying as bisexual, straight, or another way.

§Top two quartiles ("I am comfortable"/"I am very comfortable" from the question, "How would you best describe your current financial situation".

‡Includes crystal methamphetamine, mephedrone or gamma-hydroxybutyrate/gamma-butyrolactone. ¶Includes reporting of: ≥5 condomless anal sex partners, bacterial STI diagnosis, chemsex, and/or meeting partners at sex-on-premises venue, sex party, or cruising location in the last three months (i.e., since August 2023).

††Based on participant response to statement.

*See S1 Appendix for survey questions and preambles.

‡‡Ever reporting the use of antibiotics after sex for STI prevention. PEP = post-exposure prophylaxis.

doxyPEP = doxycycline post-exposure prophylaxis. HIV-PrEP = HIV pre-exposure prophylaxis. PLWHIV = people living with HIV. STI = sexually transmitted infection. SHS = sexual health service. SOP = sex-on-premises.

CAS = condomless anal sex. RiiSH = 'Reducing inequalities in Sexual Health'.

recent PrEP use or those living with HIV (PLWHIV), recent SHS use, an mpox vaccination history, 4CMenB ITU, sexual satisfaction as well as individual markers of risk (composing our composite marker) (Table 2).

In our initial multivariable model which included sociodemographic characteristics with evidence of bivariate association to antibiotic PEP use (age-group [*a priori* inclusion], country

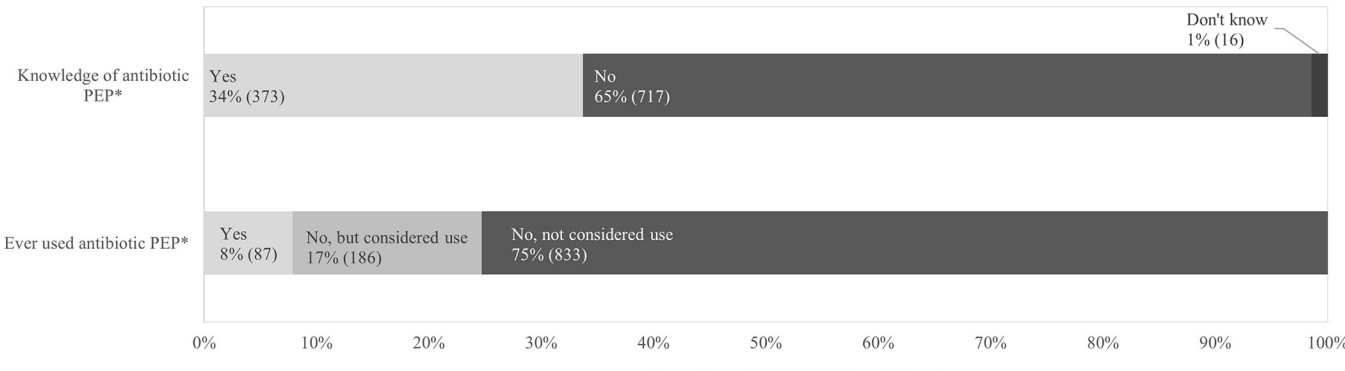

**Fig 1. Knowledge and uptake of antibiotic post-exposure prophylaxis (PEP) among RiiSH 2023 participants (n = 1,106).** *See S1 Appendix for survey questions and preambles; report of knowledge and use of antibiotics after sex for STI prevention (i.e., antibiotic PEP). PEP = post-exposure prophylaxis.

of birth, educational qualifications) and our composite marker of sexual risk, participants with lower education levels (aOR: 0.58 [95% CI: 0.34–0.97] below degree-level vs degree-level) were less likely to report use, while those born outside the UK (aOR: 1.65 [95% CI: 1.01–2.70]) and with our composite marker of risk (aOR: 2.84 [95% CI: 1.81–4.47] yes vs no) were more likely to report use.

In subsequent multivariable analyses adjusted for age-group, country of birth, educational qualifications, and our composite of sexual risk, participants reporting or considering other preventative interventions (e.g., mpox vaccination), were more likely to report STI prophylaxis use. Recent HIV-PrEP users (in the last three months) and people living with HIV (PLWHIV) were more likely to report antibiotic PEP use than HIV-negative participants not reporting recent HIV-PrEP use (in the last three months).

## Correlates of intention to use doxycycline post-exposure prophylaxis (doxyPEP ITU)

Among all participants, 51% (95% CI: 47%-56%, 568/1,106) reported doxyPEP ITU, reaching 58% (95% CI: 51%-67%, 211/361) among those with our composite marker of sexual risk (Table 3).

In bivariate analyses, sociodemographic characteristics associated with the use of doxyPEP ITU included age-group (uOR: 0.66 [95% CI: 0.46–0.95] aged 16–29 vs ≥45), which was carried forward to multivariable analyses. Those with our composite marker of sexual risk were more likely to report doxyPEP ITU (uOR: 1.53 [95% CI: 1.19–1.97] yes vs no). There were positive bivariate associations with doxyPEP ITU among those reporting recent PrEP use, PLWHIV, recent SHS use, mpox vaccination history, 4CMenB ITU, as well as individual markers of risk. While there was weak evidence supporting an association to doxyPEP ITU in those reporting high anxiety (uOR: 0.78 [95% CI: 0.61–1.00] yes vs no), we carried anxiety measures forward to multivariable modelling given previously described associations to prevention intervention uptake [29].

Those with our composite marker of sexual risk were more likely to report doxyPEP ITU (aOR: 1.50 [95% CI: 1.16–1.94]) in our initial multivariable model which included sociodemographic characteristics with evidence of bivariate association to doxyPEP ITU (age-group) and our composite marker of sexual risk. While we found no statistically significant association with age-group, there was weak evidence of decreased doxyPEP ITU in younger age-groups (aOR 0.71 [95% CI: 0.49–1.03] aged 16–29 vs ≥45 years).

**Table 2. Correlates of ever reporting antibiotic post-exposure prophylaxis (PEP) use for STI prevention among an online community sample of men and gender-diverse individuals who have sex with men taking part in the RiiSH 2023 survey.**

| | Ever used antibiotic PEP‡‡ | | | | |
|---|---|---|---|---|---|
| | row % (No.) | uOR (95% CI) | p-value | aOR (95% CI)** | p-value |
| Total | 8% (87) | | | | |
| Sociodemographic characteristics | | | | | |
| Median age at survey completion (interquartile range) | 45 (37–56) | | | | |
| Mean age at survey completion (standard deviation) | 45.7 (10.7) | | | | |
| Age-group (3 categories) | | | | | |
| 16–29 | 4% (6) | 0.43 (0.18–1.03) | | 0.53 (0.22–1.29) | |
| 30–44 | 8% (34) | 0.93 (0.59–1.48) | | 0.91 (0.57–1.48) | |
| 45 and over | 9% (47) | 1.00 (base) | 0.15 | 1.00 (base) | 0.33 |
| Gender identity and sex at birth | | | | | |
| All other gender identity groups | 7% (4) | 0.91 (0.32–2.59) | | .. | |
| Cisgender male | 8% (83) | 1.00 (base) | 0.87 | .. | |
| Sexual orientation | | | | | |
| Gay/homosexual | 8% (77) | 1.00 (base) | | .. | |
| Bisexual, straight, or another way‡ | 5% (10) | 0.58 (0.30–1.15) | 0.11 | .. | |
| Ethnic group (2 categories) | | | | | |
| All other ethnic groups | 11% (13) | 1.47 (0.79–2.73) | | .. | |
| White | 8% (74) | 1.00 (base) | 0.22 | .. | |
| Country of birth | | | | | |
| Outside of the UK | 11% (28) | 1.74 (1.09–2.80) | | 1.65 (1.01–2.70) | |
| UK | 7% (59) | 1.00 (base) | 0.020 | 1.00 (base) | 0.051 |
| Nation of residence | | | | | |
| Scotland, Wales, or N Ireland | 6% (10) | 0.72 (0.37–1.43) | | .. | |
| England | 8% (77) | 1.00 (base) | 0.35 | .. | |
| Employment | | | | | |
| Not employed | 6% (15) | 0.74 (0.42–1.32) | | .. | |
| Current employment (full-time, part-time, self-employed) | 8% (72) | 1.00 (base) | 0.31 | .. | |
| Educational qualifications | | | | | |
| Below degree-level | 5% (21) | 0.50 (0.30–0.84) | | 0.58 (0.34–0.97) | |
| Degree-level or higher | 10% (66) | 1.00 (base) | 0.007 | 1.00 (base) | 0.030 |
| Lives with partner(s) | | | | | |
| No | 8% (58) | 1.21 (0.76–1.92) | | .. | |
| Yes | 7% (29) | 1.00 (base) | 0.42 | .. | |
| Comfortable financial situation | | | | | |
| No | 7% (48) | 0.85 (0.54–1.31) | | .. | |
| Yes (top two quartiles)§ | 9% (39) | 1.00 (base) | 0.46 | .. | |
| Clinical and behavioural characteristics | | | | | |
| HIV status and HIV-PrEP use history | | | | | |
| HIV negative/unknown—No HIV-PrEP use since August 2023 | 2% (12) | 1.00 (base) | | 1.00 (base) | |
| HIV negative/unknown—HIV-PrEP use since August 2023 | 12% (53) | 6.25 (3.29–11.8) | | 4.54 (2.33–8.85) | |
| PLWHIV | 15% (22) | 7.97 (3.84–16.5) | <0.001 | 6.00 (2.81–12.8) | <0.001 |
| Recreational drug use associated with chemsex in the last year | | | | | |
| No | 7% (67) | 1.00 (base) | | .. | |
| Yes‡ | 26% (20) | 5.13 (2.91–9.05) | <0.001 | .. | |
| Bacterial STI diagnosis since August 2023 | | | | | |
| No | 7% (69) | 1.00 (base) | | .. | |

*(Continued)*

**Table 2.** (Continued)

| | Ever used antibiotic PEP‡‡ | | | | |
|---|---|---|---|---|---|
| Yes | 19% (18) | 3.15 (1.78–5.55) | <0.001 | .. | |
| SHS visit since December 2022 (last year) | | | | | |
| No | 5% (23) | 1.00 (base) | | 1.00 (base) | |
| Yes | 10% (64) | 2.17 (1.33–3.58) | 0.001 | 1.47 (0.87–2.49) | 0.14 |
| Mpox vaccination history (≥1 dose) | | | | | |
| No | 4% (28) | 1.00 (base) | | 1.00 (base) | |
| Yes | 13% (59) | 3.12 (1.95–4.97) | <0.001 | 2.24 (1.37–3.66) | 0.001 |
| "Very likely" to consider 4CMenB uptake (i.e., 4CMenB ITU) | | | | | |
| No | 4% (17) | 1.00 (base) | | 1.00 (base) | |
| Yes | 10% (70) | 2.41 (1.40–4.15) | 0.001 | 1.97 (1.12–3.46) | 0.013 |
| Partnerships since August 2023 | | | | | |
| Any new male physical sex partner(s) | | | | | |
| No new partners | 5% (19) | 0.58 (0.34–0.98) | | .. | |
| 1 or more new partners | 9% (68) | 1.00 (base) | 0.039 | .. | |
| Met male physical sex partner(s) at SOP venue, sex party, or cruising location | | | | | |
| No | 6% (60) | 1.00 (base) | | .. | |
| Yes | 17% (27) | 2.97 (1.82–4.85) | <0.001 | .. | |
| Five or more male condomless anal sex partners | | | | | |
| No | 5% (42) | 1.00 (base) | | .. | |
| Yes | 19% (45) | 4.80 (3.06–7.52) | <0.001 | .. | |
| Markers of sexual risk in the last 3 months (composite)¶ | | | | | |
| No | 5% (37) | 1.00 (base) | | 1.00 (base) | |
| Yes | 14% (65) | 3.08 (1.97–4.80) | <0.001 | 2.84 (1.81–4.47) | <0.001 |
| Personal well-being measures | | | | | |
| Low life satisfaction | | | | | |
| No | 8% (68) | 1.00 (base) | | .. | |
| Yes | 8% (19) | 0.99 (0.59–1.69) | 0.98 | .. | |
| Not specified | 0% (0) | | | | |
| Low life worthwhileness | | | | | |
| No | 7% (66) | 1.00 (base) | | .. | |
| Yes | 10% (21) | 1.33 (0.80–2.23) | 0.27 | .. | |
| Not specified | 0% (0) | | | | |
| Low happiness (yesterday) | | | | | |
| No | 8% (67) | 1.00 (base) | | .. | |
| Yes | 8% (20) | 1.02 (0.60–1.71) | 0.95 | .. | |
| Not specified | 0% (0) | | | | |
| High anxiety (yesterday) | | | | | |
| No | 9% (61) | 1.00 (base) | | .. | |
| Yes | 6% (26) | 0.71 (0.44–1.14) | 0.15 | .. | |
| Not specified | 0% (0) | | | | |
| Sexual satisfaction | | | | | |
| "I feel satisfied with my sex life"†† | | | | | |
| Disagree/do not agree nor disagree | 6% (41) | 1.00 (base) | | 1.00 (base) | |

(*Continued*)

**Table 2.** (Continued)

| | Ever used antibiotic PEP‡‡ | | | | |
|---|---|---|---|---|---|
| Agree/strongly agree | 10% (46) | 1.64 (1.06–2.54) | 0.027 | 1.36 (0.86–2.13) | 0.19 |

‡Includes those identifying as bisexual, straight, or another way.

§Top two quartiles ("I am comfortable"/"I am very comfortable" from the question, "How would you best describe your current financial situation".

‡Includes crystal methamphetamine, mephedrone or gamma-hydroxybutyrate/gamma-butyrolactone. ¶Includes reporting of: ≥5 condomless anal sex partners, bacterial STI diagnosis, chemsex, and/or meeting partners at sex-on-premises venue, sex party, or cruising location in the last three months (i.e., since August 2023).

††Based on participant response to statement. *See S1 Appendix for survey questions and preambles.

‡‡Ever reporting the use of antibiotics after sex for STI prevention (i.e., antibiotic PEP).

**Adjusted for age-group (*a priori* selection), country of birth, educational qualifications, markers of sexual risk. uOR = unadjusted odds ratio. aOR = adjusted odds ratio. PEP = post-exposure prophylaxis. HIV-PrEP = HIV pre-exposure prophylaxis. PLWHIV = people living with HIV. STI = sexually transmitted infection. SHS = sexual health service. SOP = sex-on-premises. CAS = condomless anal sex. RiiSH = 'Reducing inequalities in Sexual Health'.

In subsequent multivariable analyses adjusted for age-group and our composite marker of sexual risk, we found associations to a range of clinical and behavioural characteristics and personal well-being measures. Compared to HIV-negative, participants without recent PrEP use, PLWHIV (aOR 1.74 [95% CI: 1.28–2.57]) and HIV-negative recent PrEP users (aOR 1.40 [95% CI: 1.17–1.84]) were more likely to report doxyPEP ITU. Participants who had an SHS visit in the last year were also more likely to report doxyPEP ITU (aOR: 1.66 [95% CI: 1.29–2.14]) while those reporting high anxiety were less likely (aOR 0.77 [95% CI: 0.80–0.99]). 4CMenB ITU was highly correlated with doxyPEP ITU (aOR 8.82 [95% CI: 6.62–11.9]) (Table 3).

## Correlates of intention to use 4CMenB (4CMenB ITU)

Over two-thirds of all participants (64% [95% CI: 60%-69%], 713/1,106) reported 4CMenB ITU, increasing to 75% (95% CI: 66%-84%, 270/361) among those with markers of sexual risk in the last three months (Table 4).

In bivariate analyses, sociodemographic characteristics associated with 4CMenB ITU included age-group (uOR: 1.60 [95% CI: 1.09–2.35] aged 16–29 vs ≥45), sexual orientation (uOR: 0.62 [95% CI: 0.45–0.85] bisexual, straight, another way vs gay/homosexual), employment (0.74 [95% CI: 0.55–0.99] yes vs no), and educational qualifications (uOR: 0.54 [95% CI: 0.42–0.69] below degree-level vs degree-level) and were carried forward to multivariable analyses. There were positive bivariate associations with 4CMenB ITU among those reporting recent PrEP use, PLWHIV, recent SHS use, mpox vaccination history, antibiotic PEP use, doxyPEP ITU, as well as individual markers of risk.

In our initial multivariable model which included all sociodemographic characteristics with evidence of bivariate association to 4CMenB ITU (age-group, sexual orientation, employment, educational qualifications) and our composite marker of sexual risk, younger participants were more likely to report 4CMenB ITU (aOR 2.16 [95% CI: 1.44–3.25] aged 16–29 vs ≥45 years), while bisexual or straight-identifying participants (aOR 0.62 [95% CI: 0.45–0.86]) and those with lower educational qualifications (aOR 0.57 [95% CI: 0.44–0.73]) were less likely to report 4CMenB ITU. Participants with our composite marker of sexual risk were more than twice as likely to report ITU (aOR 2.09 [95% CI: 1.56–2.79]).

In subsequent multivariable analyses, adjusted for age-group, sexual orientation, employment, educational qualifications, and our composite marker of sexual risk, PLWHIV (aOR: 1.96 [95% CI: 1.28–2.99]) and recent HIV-negative PrEP users (aOR: 2.26 [95% CI: 1.67–3.05]) were about twice as likely to report 4CMenB ITU compared to HIV-negative

**Table 3. Correlates of reporting intention to use doxyPEP (doxyPEP ITU) among an online community sample of men and gender-diverse individuals who have sex with men taking part in the RiiSH 2023 survey.**

| | Reporting intention to use doxyPEP (i.e., doxyPEP ITU)‡‡ | | | | |
|---|---|---|---|---|---|
| | row % (No.) | uOR (95% CI) | p-value | aOR (95% CI)** | p-value |
| Total | 51% (568) | | | | |
| Sociodemographic characteristics | | | | | |
| Median age at survey completion (interquartile range) | 44 (35–54) | | | | |
| Mean age at survey completion (standard deviation) | 44.4 (12.3) | | | | |
| Age-group (3 categories) | | | | | |
| 16–29 | 42% (63) | 0.66 (0.46–0.95) | | 0.71 (0.49–1.03) | |
| 30–44 | 54% (225) | 1.09 (0.84–1.41) | | 1.13 (0.87–1.46) | |
| 45 and over | 52% (280) | 1.00 (base) | 0.031 | 1.00 (base) | 0.059 |
| Gender identity and sex at birth | | | | | |
| All other gender identity groups | 45% (25) | 0.78 (0.45–1.34) | | .. | |
| Cisgender male | 52% (543) | 1.00 (base) | 0.37 | .. | |
| Sexual orientation | | | | | |
| Gay/homosexual | 51% (467) | 1.00 (base) | | .. | |
| Bisexual, straight, or another way‡ | 52% (101) | 1.01 (0.74–1.37) | 0.96 | .. | |
| Ethnic group (2 categories) | | | | | |
| All other ethnic groups | 53% (65) | 1.09 (0.75–1.59) | | .. | |
| White | 51% (503) | 1.00 (base) | 0.65 | .. | |
| Country of birth | | | | | |
| Outside of the UK | 54% (134) | 1.17 (0.88–1.56) | | .. | |
| UK | 50% (434) | 1.00 (base) | 0.27 | .. | |
| Nation of residence | | | | | |
| Scotland, Wales, or N Ireland | 52% (85) | 1.01 (0.72–1.40) | | .. | |
| England | 51% (483) | 1.00 (base) | 0.96 | .. | |
| Employment | | | | | |
| Not employed | 47% (113) | 0.82 (0.62–1.09) | | .. | |
| Current employment (full-time, part-time, self-employed) | 52% (455) | 1.00 (base) | 0.18 | .. | |
| Educational qualifications | | | | | |
| Below degree-level | 52% (215) | 1.03 (0.81–1.31) | | .. | |
| Degree-level or higher | 51% (353) | 1.00 (base) | 0.82 | .. | |
| Lives with partner(s) | | | | | |
| No | 49% (343) | 0.82 (0.64–1.05) | | .. | |
| Yes | 54% (225) | 1.00 (base) | 0.11 | .. | |
| Comfortable financial situation | | | | | |
| No | 50% (328) | 0.90 (0.71–1.15) | | .. | |
| Yes (top two quartiles)§ | 53% (240) | 1.00 (base) | 0.40 | .. | |
| Clinical and behavioural characteristics | | | | | |

*(Continued)*

**Table 3.** (Continued)

| | Reporting intention to use doxyPEP (i.e., doxyPEP ITU)‡‡ | | | | |
|---|---|---|---|---|---|
| HIV status and HIV-PrEP use history | | | | | |
| HIV negative/unknown–No HIV-PrEP use since August 2023 | 45% (242) | 1.00 (base) | | 1.00 (base) | |
| HIV negative/unknown–HIV-PrEP use since August 2023 | 56% (238) | 1.56 (1.21–2.01) | | 1.40 (1.17–1.84) | |
| PLWHIV | 21% (88) | 1.96 (1.34–2.85) | <0.001 | 1.74 (1.28–2.57) | 0.006 |
| Recreational drug use associated with chemsex in the last year | | | | | |
| No | 51% (524) | 1.00 (base) | | .. | |
| Yes‡ | 58% (44) | 1.33 (0.83–2.13) | 0.24 | .. | |
| Bacterial STI diagnosis since August 2023 | | | | | |
| No | 50% (501) | 1.00 (base) | | .. | |
| Yes | 70% (67) | 2.35 (1.49–3.69) | <0.001 | .. | |
| SHS visit since December 2022 (last year) | | | | | |
| No | 43% (202) | 1.00 (base) | | 1.00 (base) | |
| Yes | 58% (366) | 1.80 (1.41–2.29) | <0.001 | 1.66 (1.29–2.14) | <0.001 |
| Mpox vaccination history (≥1 dose) | | | | | |
| No | 48% (303) | 1.00 (base) | | 1.00 (base) | |
| Yes | 56% (265) | 1.42 (1.12–1.81) | 0.004 | 1.28 (1.10–1.64) | 0.054 |
| "Very likely" to consider 4CmenB uptake (i.e., 4CmenB ITU) | | | | | |
| No | 21% (81) | 1.00 (base) | | 1.00 (base) | |
| Yes | 68% (487) | 8.30 (6.21–11.1) | <0.001 | 8.82 (6.62–11.93) | <0.001 |
| Partnerships since August 2023 | | | | | |
| Any new male physical sex partner(s) | | | | | |
| No new partners | 46% (162) | 0.74 (0.57–0.95) | | .. | |
| 1 or more new partners | 54% (406) | 1.00 (base) | 0.018 | .. | |
| Met male physical sex partner(s) at SOP venue, sex party, or cruising location | | | | | |
| No | 51% (481) | 1.00 (base) | | .. | |
| Yes | 54% (87) | 1.13 (0.81–1.59) | 0.46 | .. | |
| Five or more male condomless anal sex partners | | | | | |
| No | 48% (416) | 1.00 (base) | | .. | |
| Yes | 66% (152) | 2.12 (1.57–2.87) | <0.001 | .. | |
| Markers of sexual risk in the last 3 months (composite)¶ | | | | | |
| No | 48% (357) | 1.00 (base) | | 1.00 (base) | |
| Yes | 58% (211) | 1.53 (1.19–1.97) | 0.001 | 1.50 (1.16–1.94) | <0.001 |
| Personal well-being measures | | | | | |
| Low life satisfaction | | | | | |
| No | 52% (445) | 1.00 (base) | | .. | |
| Yes | 50% (121) | 0.94 (0.70–1.25) | 0.35 | .. | |

(*Continued*)

**Table 3.** (Continued)

| | Reporting intention to use doxyPEP (i.e., doxyPEP ITU)‡‡ | | | | |
|---|---|---|---|---|---|
| Not specified | 100% (2) | | | | |
| Low life worthwhileness | | | | | |
| No | 51% (453) | 1.00 (base) | | .. | |
| Yes | 52% (113) | 1.04 (0.77–1.40) | 0.37 | .. | |
| Not specified | 100% (2) | | | | |
| Low happiness (yesterday) | | | | | |
| No | 51% (435) | 1.00 (base) | | .. | |
| Yes | 52% (131) | 1.05 (0.79–1.39) | 0.37 | .. | |
| Not specified | 100% (2) | | | | |
| High anxiety (yesterday) | | | | | |
| No | 54% (372) | 1.00 | | 1.00 (1.00 | |
| Yes | 47% (194) | 0.78 (0.61–1.00) | 0.057 | 0.77 (0.80–0.99) | 0.040 |
| Not specified | 100% (2) | | | | |
| Sexual satisfaction | | | | | |
| "I feel satisfied with my sex life"†† | | | | | |
| Disagree/do not agree nor disagree | 51% (327) | 1.00 | | .. | |
| Agree/strongly agree | 52% (241) | 1.07 (0.88–1.20) | 0.56 | .. | |

‡Includes those identifying as bisexual, straight, or another way.

§Top two quartiles ("I am comfortable"/ "I am very comfortable" from the question, "How would you best describe your current financial situation".

‡Includes crystal methamphetamine, mephedrone or gamma-hydroxybutyrate/gamma-butyrolactone. ¶Includes reporting of: ≥5 condomless anal sex partners, bacterial STI diagnosis, chemsex, and/or meeting partners at sex-on-premises venue, sex party, or cruising location in the last three months (i.e., since August 2023).

††Based on participant response to statement. *See S1 Appendix for survey questions and preambles. ‡‡Participants 'very likely' to consider doxyPEP uptake (See S1 Appendix for survey questions and preambles).

**Adjusted for age-group (*a priori* selection), markers of sexual risk. uOR = unadjusted odds ratio. aOR = adjusted odds ratio. PEP = post-exposure prophylaxis. doxyPEP = doxycycline post-exposure prophylaxis. HIV-PrEP = HIV pre-exposure prophylaxis. PLWHIV = people living with HIV. STI = sexually transmitted infection. SHS = sexual health service. SOP = sex-on-premises. CAS = condomless anal sex. RiiSH = 'Reducing inequalities in Sexual Health'.

participants not reporting recent HIV-PrEP use. Sexual health service use in the last year (aOR 2.45 [95% CI: 1.87–3.21]) and uptake of preventative interventions were positively associated with 4CMenB ITU, and there was high correlation with doxyPEP ITU (aOR 10.1 [95% CI: 7.33–13.8]). Those who ever used antibiotic PEP were twice as likely to report 4CMenB ITU (aOR: 1.99 [95% CI: 1.13–3.50]) (Table 4).

## Discussion

We show that the majority of men and gender-diverse individuals having sex with men in our community sample would choose to access doxyPEP and the 4CMenB vaccine were they available for use in the UK at SHS, and that intention to use is greater in those potentially most likely to benefit. Over half of participants (51%) expressed an intention to use doxyPEP, with even greater levels (64%) reporting intention to use 4CMenB. While findings demonstrate substantial interest in the use of doxyPEP, fewer than one in ten (8%) participants who responded to this survey reported use of antibiotic PEP, with usage more common among those at greater risk of STIs. These findings, however, are based on small absolute numbers, but update

**Table 4. Correlates of reporting intention to use 4CMenB (4CMenB ITU) among an online community sample of men and gender-diverse individuals who have sex with men taking part in the RiiSH 2023 survey.**

| | Reporting intention to use 4CMenB (i.e., 4CMenB ITU)¶¶ | | | | |
|---|---|---|---|---|---|
| | row % (No.) | uOR (95% CI) | p-value | aOR (95% CI)** | p-value |
| Total | 64% (713) | | | | |
| Sociodemographic characteristics | | | | | |
| Median age at survey completion (interquartile range) | 42 (34–52) | | | | |
| Mean age at survey completion (standard deviation) | 42.8 (12.2) | | | | |
| Age-group (3 categories) | | | | | |
| 16–29 | 69% (104) | 1.60 (1.09–2.35) | | 2.16 (1.44–3.25) | |
| 30–44 | 71% (296) | 1.78 (1.36–2.34) | | 1.84 (1.39–2.45) | |
| 45 and over | 58% (313) | 1.00 (base) | <0.001 | 1.00 (base) | <0.001 |
| Gender identity and sex at birth | | | | | |
| All other gender identity groups | 67% (37) | 1.14 (0.64–2.03) | | .. | |
| Cisgender male | 64% (676) | 1.00 (base) | 0.66 | .. | |
| Sexual orientation | | | | | |
| Gay/homosexual | 66% (605) | 1.00 (base) | | 1.00 (base) | |
| Bisexual, straight, or another way‡ | 55% (108) | 0.62 (0.45–0.85) | 0.003 | 0.62 (0.45–0.86) | 0.0044 |
| Ethnic group (2 categories) | | | | | |
| All other ethnic groups | 66% (80) | 1.06 (0.71–1.57) | | .. | |
| White | 64% (633) | 1.00 (base) | 0.79 | .. | |
| Country of birth | | | | | |
| Outside of the UK | 69% (170) | 1.31 (0.96–1.77) | | .. | |
| UK | 63% (543) | 1.00 (base) | 0.085 | .. | |
| Nation of residence | | | | | |
| Scotland, Wales, or N Ireland | 65% (107) | 1.02 (0.72–1.44) | | .. | |
| England | 64% (606) | 1.00 (base) | 0.91 | .. | |
| Employment | | | | | |
| Not employed | 59% (140) | 0.74 (0.55–0.99) | | 0.85 (0.62–1.15) | |
| Current employment (full-time, part-time, self-employed) | 66% (573) | 1.00 (base) | 0.040 | 1.00 (base) | 0.29 |
| Educational qualifications | | | | | |
| Below degree-level | 55% (230) | 0.54 (0.42–0.69) | | 0.57 (0.44–0.73) | |
| Degree-level or higher | 70% (483) | 1.00 (base) | <0.001 | 1.00 (base) | <0.001 |
| Lives with partner(s) | | | | | |
| No | 65% (448) | 1.02 (0.79–1.32) | | .. | |
| Yes | 64% (265) | 1.00 (base) | 0.87 | .. | |
| Comfortable financial situation | | | | | |
| No | 63% (411) | 0.86 (0.67–1.10) | | .. | |
| Yes (top two quartiles)§ | 67% (302) | 1.00 (base) | 0.23 | .. | |
| Clinical and behavioural characteristics | | | | | |

*(Continued)*

**Table 4.** (Continued)

| | Reporting intention to use 4CMenB (i.e., 4CMenB ITU)¶¶ | | | | |
|---|---|---|---|---|---|
| HIV status and HIV-PrEP use history | | | | | |
| HIV negative/unknown—No HIV-PrEP use since August 2023 | 54% (289) | 1.00 (base) | | 1.00 (base) | |
| HIV negative/unknown—HIV-PrEP use since August 2023 | 76% (323) | 2.73 (2.06–3.61) | | 2.26 (1.67–3.05) | |
| PLWHIV | 71% (101) | 2.07 (1.39–3.08) | <0.001 | 1.96 (1.28–2.99) | <0.001 |
| Recreational drug use associated with chemsex in the last year | | | | | |
| No | 63% (652) | 1.00 (base) | | .. | |
| Yes‡ | 80% (61) | 2.36 (1.32–4.21) | 0.003 | .. | |
| Bacterial STI diagnosis since August 2023 | | | | | |
| No | 63% (635) | 1.00 (base) | | .. | |
| Yes | 81% (78) | 2.56 (1.51–4.34) | <0.001 | .. | |
| SHS visit since December 2022 (last year) | | | | | |
| No | 50% (236) | 1.00 (base) | | 1.00 (base) | |
| Yes | 75% (477) | 2.97 (2.30–3.84) | <0.001 | 2.45 (1.87–3.21) | <0.001 |
| Mpox vaccination history (≥1 dose) | | | | | |
| No | 55% (347) | 1.00 (base) | | 1.00 (base) | |
| Yes | 78% (366) | 2.93 (2.24–3.83) | <0.001 | 2.38 (1.79–3.16) | <0.001 |
| "Very likely" to consider doxyPEP uptake (i.e., doxyPEP ITU) | | | | | |
| No | 32% (226) | 1.00 (base) | | 1.00 (base) | |
| Yes | 68% (487) | 8.30 (6.21–11.10) | <0.001 | 10.1 (7.33–13.8) | <0.001 |
| Ever used antibiotic PEP‡‡ | | | | | |
| No | 63% (643) | 1.00 (base) | | 1.00 (base) | |
| Yes | 80% (70) | 2.41 (1.40–4.15) | <0.001 | 1.99 (1.13–3.50) | 0.013 |
| Partnerships since August 2023 | | | | | |
| Any new male physical sex partner(s) | | | | | |
| No new partners | 54% (188) | 0.51 (0.39–0.66) | | .. | |
| 1 or more new partners | 70% (525) | 1.00 (base) | <0.001 | .. | |
| Met male physical sex partner(s) at SOP venue, sex party, or cruising location | | | | | |
| No | 54% (188) | 1.00 (base) | | .. | |
| Yes | 70% (525) | 1.26 (0.88–1.81) | <0.001 | .. | |
| Five or more male condomless anal sex partners | | | | | |
| No | 60% (524) | 1.00 (base) | | .. | |
| Yes | 82% (189) | 3.01 (2.10–4.32) | <0.001 | .. | |
| Markers of sexual risk in the last 3 months (composite)¶ | | | | | |
| No | 59% (443) | 1.00 (base) | | 1.00 (base) | |
| Yes | 75% (270) | 2.02 (1.53–2.67) | <0.001 | 2.09 (1.56–2.79) | <0.001 |
| Personal well-being measures | | | | | |

(Continued)

**Table 4.** (Continued)

| | Reporting intention to use 4CMenB (i.e., 4CMenB ITU)¶¶ | | | | |
|---|---|---|---|---|---|
| Low life satisfaction | | | | | |
| No | 65% (560) | 1.00 (base) | | .. | |
| Yes | 63% (152) | 0.91 (0.68–1.22) | 0.75 | .. | |
| Not specified | 50% (1) | | | | |
| Low life worthwhileness | | | | | |
| No | 65% (579) | 1.00 (base) | | .. | |
| Yes | 61% (133) | 0.84 (0.62–1.14) | 0.50 | .. | |
| Not specified | 50% (1) | | | | |
| Low happiness (yesterday) | | | | | |
| No | 65% (551) | 1.00 (base) | | .. | |
| Yes | 64% (161) | 0.98 (0.73–1.32) | 0.90 | .. | |
| Not specified | 50% (1) | | | | |
| High anxiety (yesterday) | | | | | |
| No | 64% (442) | 1.00 (base) | | .. | |
| Yes | 66% (270) | 1.11 (0.86–1.44) | 0.66 | .. | |
| Not specified | 50% (1) | | | | |
| Sexual satisfaction | | | | | |
| "I feel satisfied with my sex life"†† | | | | | |
| Disagree/do not agree nor disagree | 63% (404) | 1.00 (base) | | .. | |
| Agree/strongly agree | 67% (309) | 1.23 (0.95–1.58) | 0.11 | .. | |

‡Includes those identifying as bisexual, straight, or another way.

§Top two quartiles ("I am comfortable"/"I am very comfortable" from the question, "How would you best describe your current financial situation".

‡Includes crystal methamphetamine, mephedrone or gamma-hydroxybutyrate/gamma-butyrolactone. ‡‡Ever reporting the use of antibiotics after sex for STI prevention (i.e., antibiotic PEP). ¶Includes reporting of: ≥5 condomless anal sex partners, bacterial STI diagnosis, chemsex, and/or meeting partners at sex-on-premises venue, sex party, or cruising location in the last three months (i.e., since August 2023). ††Based on participant response to statement.

*See S1 Appendix for survey questions and preambles.

¶¶Participants 'very likely' to consider 4CMenB uptake (See S1 Appendix for survey questions and preambles).

**Adjusted for age-group (*a priori* selection), sexual orientation, employment status, educational qualifications, markers of sexual risk. uOR = unadjusted odds ratio. aOR = adjusted odds ratio. PEP = post-exposure prophylaxis. doxyPEP = doxycycline post-exposure prophylaxis. HIV-PrEP = HIV pre-exposure prophylaxis. PLWHIV = people living with HIV. STI = sexually transmitted infection. SHS = sexual health service. SOP = sex-on-premises. CAS = condomless anal sex. RiiSH = 'Reducing inequalities in Sexual Health'.

previous estimates of the extent of antibiotic PEP use in men and gender-diverse individuals having sex with men prior to the publication of the first UK guidelines.

Among those using antibiotic PEP, most reported use of appropriate, evidence-based antibiotics for use as post-exposure STI prophylaxis (i.e., doxycycline). However, there were indications of unknown and inappropriate antibiotic use, which is concerning given this may drive antimicrobial resistance (AMR) as well as cause individual harm, possibly to a greater extent than doxyPEP use. Over a third of all participants reported knowledge of antibiotic PEP for STI prevention, and while few reported uptake, one in five participants who had not reported use had ever considered taking antibiotics to prevent STIs.

Consistent with previous studies [10–12], we found higher uptake of STI prophylaxis in PLWHIV (15%), HIV-PrEP users (12%), and participants at greater risk of STIs as indicated by recent markers of sexual risk (14%). Participants with lower levels of education were less likely to report antibiotic PEP use, while prior mpox vaccination uptake was positively associated with use. Compared with a recent study in Germany which estimated doxyPEP uptake in around 20% of GBMSM [30], our sample estimates were lower but this may reflect the wider availability of other STI preventative interventions in the UK through free and confidential SHS [31, 32]. Differing uptake could also reflect unavailability of doxyPEP in UK SHS. Given the current evidence base, we framed uptake of prophylactic antibiotics around post-exposure use. Questions across behavioural surveys, including prior RiiSH rounds, have included questions about both pre- and post-exposure use. This may limit comparisons with estimates in previous literature and underestimate overall use of STI prophylaxis as STI prevention. Future research, monitoring and evaluation would benefit from a consistent definition, which will itself be facilitated by the upcoming national UK guidelines.

Over half of participants indicated doxyPEP ITU. While there was a high correlation between doxyPEP and 4CMenB ITU, a higher proportion of participants—approximately two-thirds—reported 4CMenB ITU. AMR concerns, consistently highlighted in contemporary health settings as part of antimicrobial stewardship initiatives [33], could explain lower doxyPEP ITU. Those reporting high anxiety were less likely to report doxyPEP ITU, which may signal hesitancy arising from AMR worries and reluctance to use antibiotics prophylactically. Similar views were found in a qualitative study of SHS attendees, where there was greater acceptability and support of vaccinations for STI prevention as alternatives to antibiotic prophylaxis use given concerns about side effects and safety [34]. We found levels of 4CMenB ITU were similar to the high self-reported uptake of opportunistically offered Hepatitis A, Hepatitis B and human papillomavirus (HPV) vaccination in SHS in similar online community samples [35], which may influence greater 4CMenB uptake acceptability.

As with antibiotic PEP uptake, PLWHIV, recent HIV-PrEP users, those accessing SHS, and those at greater sexual risk were more likely to report 4CMenB ITU. However, we also found further associations with age and sexual orientation. Younger age-groups were more likely to report 4CMenB ITU, which may present opportunities for further prevention intervention education and uptake in groups with known STI outcome inequalities [36]. Like recent vaccination uptake examinations in prior RiiSH surveys [35, 37], we found differences by socioeconomic characteristics, with lower 4CMenB ITU in those reporting educational qualifications below degree-level, as well as lower ITU among sexual minorities (i.e., bisexual and straight-identifying participants). Inclusive and accessible health promotion and patient education surrounding novel STI prevention interventions, especially considering self-sourcing of antibiotics, will be imperative to minimise knowledge and uptake inequalities and to clearly inform potential, though yet unclear, risks in balance of benefits to STI prophylaxis use.

## Limitations

This cross-sectional study is subject to limitations. Compared to the most recent Natsal survey (Natsal-3), a national probability survey representative of Great Britain, our survey participants report higher educational attainment and levels of employment, potentially indicating higher health literacy relative to the general GBMSM population [38]. Participants likely experience higher levels of sexual risk relative to the wider population in the UK, as GBMSM taking part in targeted behavioural surveys consistently reported more sexual risk behaviours relative to nationally representative samples [39]; alongside, the use of the internet and social networking in finding sexual partners has been associated with the report of markers of sexual risk

[40]. While we aimed to recruit from a broader spectrum of men and gender-diverse individuals having sex with men who likely have differing sexual risk relative to clinic recruited samples, participants likely represent key populations targeted for HIV and STI prevention interventions. Aggregate census data available by sexual orientation and gender identity, respectively, limit comparisons of our participants to the wider population of men and gender-diverse individuals having sex with men in the UK [41, 42].

ITU measures may not equate to uptake in practice but help gauge interest and acceptability among participants who would likely comprise targeted groups for these interventions. Dichotomising outcome measures limits interpretation and could have led to conservative ITU estimates. While we posited that sexual satisfaction could be influenced by actual or intended uptake of STI prevention interventions, we found no evidence of association, though larger studies are needed to examine the role of uptake on more holistic measures of sexual satisfaction and well-being. We collected no personal identifiers to facilitate the reporting of sensitive data on sexual behaviours; however, survey responses may be subject to social acceptability bias. Sample selection could be subject to digital exclusion, and we do not know how representative our study sample is of men and gender-diverse individuals having sex with men using social networking and dating applications. However, consistent methodology and community-partnered implementation are strengths of this long-running survey series and consistently aid the characterisation of behavioural risk, currently absent from available national STI surveillance, as well as actual and intended preventative intervention uptake among key populations in the UK.

## Implications

Findings from this study illustrate a community sample of men and gender-diverse individuals having sex with men with considerable interest in the use of novel biomedical STI interventions, which is not dissimilar to the high uptake seen of HIV-PrEP and other STI and viral hepatitis vaccination in UK among key groups. As seen in the PrEP Impact Trial, an implementation trial assessing HIV-PrEP eligibility and uptake in the UK from 2017–2020 [43], eligibility for trial places quickly outstripped availability at the onset of the trial (10,000, increasing to 26,000), largely attributable to early knowledge and engagement facilitated by community organisations [44, 45]. A significant proportion of our community sample indicated likely uptake of these preventative interventions with high correlation of ITU for both, signalling the benefit of shared education and offer of prevention interventions within SHS or outreach settings. Participants at greater risk of STIs were more likely to report ever using STI prophylaxis as well as doxyPEP and 4CMenB ITU, suggesting an appropriate assessment of personal sexual risk, however, we did find potential sociodemographic and SHS engagement-based uptake differences common in uptake of other preventative interventions such as mpox vaccination and HIV-PrEP.

Given limited empirical evidence [13, 15], it is unclear whether doxyPEP will increase the development of AMR in STIs and non-STI bacteria. However, this is a key area of concern, and a focus for surveillance and research. While there is no recommendation for the use of doxyPEP as STI prevention in the UK at the time of writing, there are early indications of knowledge of antibiotic PEP, as well as uptake through private or off-license purchasing. Community-led knowledge mobilisation efforts [46] have likely boosted awareness of evidence-based uptake regimens associated with STI incidence declines in GBMSM, now recommended in the United States [47, 48]. At present, the upcoming UK doxyPEP guidelines will likely influence the decision of implementation across SHS. Also, the JCVI recommendation for a targeted opportunistic gonorrhoea vaccination programme using 4CMenB is awaiting

approval by health ministers in the UK. As we found few studies exploring acceptability of both doxyPEP or 4CMenB among men and gender-diverse individuals having sex with men, we suggest further studies investigating preferred prevention interventions to identify knowledge gaps and health concerns that may influence uptake. While implementation of both doxyPEP and 4CMenB vaccination would likely be through free and confidential SHS in the UK, this may not be the case in other countries. Given the considerable interest in doxyPEP and 4CMenB in the UK, packaging education and outreach that includes a range of preventative interventions will be key in optimising sexual health related contacts and ensuring informed use in key populations.

## Conclusion

While regular testing, condom promotion, and treatment as prevention remain an integral foundation to STI prevention in the UK, these approaches have not curbed rapid increases in STI diagnoses among GBMSM and other key populations in the last decade. Novel prevention interventions should be considered to supplement existing control strategies. As adoption of doxyPEP, 4CMenB vaccination and other interventions across SHS is considered in the UK, future guidelines and health promotion messaging must be carefully crafted alongside clinical experts and community partners given intervention complexity, implications to sexual behaviour and AMR, and the risk of presenting conflicting public health messages regarding antibiotic stewardship. Robust monitoring and evaluation will be crucial to understand the impact of doxyPEP and 4CMenB use on AMR and STI incidence in key populations if rolled out across SHS. Given lessons learned from the implementation of HIV prevention interventions, the magnitude of community interest and uptake must not be underestimated and there will be a need to ensure equitable access to these interventions to those in greatest need [24]. There will be a similar need for equity considerations in health promotion, patient education, and uptake across different service models, including those delivered primarily online, depending on local need. Minimising knowledge and potential accessibility barriers for those who may benefit from future STI preventative interventions must be considered from the outset of planning and embedded in implementation and monitoring to limit health inequalities and to facilitate empowered decision-making that benefits individual-level sexual wellbeing and autonomy.

## Supporting information

**S1 Appendix. Excerpt of RiiSH 2023 survey questions and preambles used for study outcome measures.**
(DOCX)

**S2 Appendix. RiiSH 2023 participant flowchart.**
(DOCX)

**S3 Appendix. Antibiotic regimens ever used as antibiotic post-exposure prophylaxis (PEP) among RiiSH 2023 participants.**
(DOCX)

**S4 Appendix. Age and ethnic group among RiiSH 2023 participants by recruitment site.**
(DOCX)

## Acknowledgments

We thank Elizabeth Fearon (University College London), Takudzwa Mukiwa (Terrence Higgins Trust), Will Nutland (The Love Tank) and Benjamin Weil (The Love Tank) for their

review and contributions to the RiiSH 2023 survey. We thank Emmanuel Musah for review of our draft manuscript.

We acknowledge members of the National Institute for Health and Care Research Health Protection Research Unit (NIHR HPRU) in Blood Borne and Sexually Transmitted Infections (BBSTI) Steering Committee: Caroline Sabin, John Saunders, Catherine H. Mercer, Hamish Mohammed (previously Gwenda Hughes), Greta Rait, Ruth Simmons, William Rosenberg, Tamyo Mbisa, Rosalind Raine, Sema Mandal, Rosamund Yu, Samreen Ijaz, Fabiana Lorencatto, Rachel Hunter, Kirsty Foster and Mamooma Tahir.

## Author Contributions

**Conceptualization:** Dana Ogaz, Jessica Edney, Dawn Phillips, Dolores Mullen, David Reid, Ruth Wilkie, Erna Buitendam, James Bell, Catherine M. Lowndes, Gwenda Hughes, Catherine H. Mercer, John Saunders, Hamish Mohammed.

**Data curation:** Dana Ogaz, Jessica Edney, Ruth Wilkie, Catherine H. Mercer, Hamish Mohammed.

**Formal analysis:** Dana Ogaz.

**Methodology:** Dana Ogaz, Dolores Mullen, David Reid, Ruth Wilkie, Gwenda Hughes, Catherine H. Mercer, John Saunders, Hamish Mohammed.

**Project administration:** Dawn Phillips.

**Supervision:** Gwenda Hughes, Helen Fifer, Catherine H. Mercer, John Saunders, Hamish Mohammed.

**Validation:** Jessica Edney, Dolores Mullen.

**Visualization:** Dana Ogaz.

**Writing – original draft:** Dana Ogaz.

**Writing – review & editing:** Dana Ogaz, Jessica Edney, Dawn Phillips, Dolores Mullen, David Reid, Ruth Wilkie, Erna Buitendam, James Bell, Catherine M. Lowndes, Gwenda Hughes, Helen Fifer, Catherine H. Mercer, John Saunders, Hamish Mohammed.

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
