## [Decision Letter · Decision Letter 0]

11 Jul 2024

PGPH-D-24-01329

Knowledge, uptake and intention to use antibiotic post-exposure prophylaxis and meningococcal B vaccine (4CMenB) for gonorrhoea among a large, online community sample of gay, bisexual and other men who have sex with men in the UK

Dear Dr. Ogaz,

Thank you for submitting your manuscript to PLOS Global Public Health. After careful consideration, we feel that it has merit but does not fully meet PLOS Global Public Health’s publication criteria as it currently stands. Therefore, we invite you to submit a revised version of the manuscript that addresses the points raised during the review process.

This paper addresses an important topic in shaping the future of biomedical STI prevention. While it is in general well-written, the reviewers provided valuable inputs for improving the manuscript. 

We look forward to receiving your revised manuscript.

Kind regards,

Tsz Ho Kwan, PhD

Academic Editor

Journal Requirements:

1. In your Methods section, please include additional information about your dataset and ensure that you have included a statement specifying whether the collection and analysis method complied with the terms and conditions for the source of the data.

a. Please clarify all sources of funding (financial or material support) for your study. List the grants (with grant number) or organizations (with url) that supported your study, including funding received from your institution. 

b. State the initials, alongside each funding source, of each author to receive each grant.

c. State what role the funders took in the study. If the funders had no role in your study, please state: “The funders had no role in study design, data collection and analysis, decision to publish, or preparation of the manuscript.”

If you did not receive any funding for this study, please simply state: “The authors received no specific funding for this work.”"

3. In the online submission form you indicate that your data is not available for proprietary reasons and have provided a contact point for accessing this data. Please note that your current contact point is a co-author on this manuscript. According to our Data Policy, the contact point must not be an author on the manuscript and must be an institutional contact, ideally not an individual. Please revise your data statement to a non-author institutional point of contact, such as a data access or ethics committee, and send this to us via return email. Please also include contact information for the third party organization, and please include the full citation of where the data can be found.

4. Please provide separate figure files in .tif or .eps format.

5. Tables should not be uploaded as individual files. Please remove these files and include the Tables in your manuscript file as editable, cell-based objects. For more information about how to format tables, see our guidelines: 

https://journals.plos.org/globalpublichealth/s/tables

Additional Editor Comments (if provided):

Reviewers' comments:

Reviewer's Responses to Questions

**Comments to the Author**

1. Does this manuscript meet PLOS Global Public Health’s publication criteria? Is the manuscript technically sound, and do the data support the conclusions? The manuscript must describe methodologically and ethically rigorous research with conclusions that are appropriately drawn based on the data presented.

Reviewer #1: Yes

Reviewer #2: Yes

Reviewer #3: Yes

2. Has the statistical analysis been performed appropriately and rigorously?

Reviewer #1: Yes

Reviewer #2: No

Reviewer #3: Yes

3. Have the authors made all data underlying the findings in their manuscript fully available (please refer to the Data Availability Statement at the start of the manuscript PDF file)?

Reviewer #1: No

Reviewer #2: Yes

Reviewer #3: Yes

4. Is the manuscript presented in an intelligible fashion and written in standard English?

Reviewer #1: Yes

Reviewer #2: Yes

Reviewer #3: Yes

5. Review Comments to the Author

Reviewer #1: This is a clearly written manuscript with compelling findings. However I have several comments and suggestions regarding the generalizability of the findings to the overall target population.

Overall

1) While the authors do mention limitations of the recruitment method, particularly the possibility of digital exclusion, this is a major concern with regard to potential generalizability of these findings. It would be helpful to provide some context for how representative this sample is of what is known about GBMSM in the UK, for example with regard to education level, race, etc.

2) Additionally, were demographics of the participants recruited through the community-cascaded link different from the remainder of the sample?

3) In addition to concerns about AMR mentioned in the discussion, are there other potential reasons why doxyPEP ITU may be lower than 4CMenB ITU? Eg, concerns about side effects, need for adherence to dosing/timing recommendations, etc.

4) This manuscript is understandably heavily focused on the UK, which does have a unique national health system that may influence both the findings and their immedicate implications. However are there implications of these findings that can be extrapolated to doxyPEP and 4CMenB public health messaging and programming outside the UK? This would be helpful for readers of a global public health journal.

Minor text comments:

Lines 107-108 “online click-throughs were not captured” – please explain this for readers not familiar with data collection through online surveys

Lines 198-199: “reaching 68% among those with markers of sexual risk” – in the table this appears to be 58%

Reviewer #2: In their manuscript, Ogaz et al. examine the knowledge, uptake and intention to use antibiotic post-exposure prophylaxis and meningococcal B vaccine for N. gonorrhoea infection among GBMSM in the UK. They observed that nearly half of respondents were highly likely to take up either strategy. The research question is of interest; however, there are certain issues that need to be considered.

There are three outcomes studied and the link between the three are not entirely obvious. One involves the actual use of antibiotics for PEP (which is not exclusively doxyPEP). The others involve intent to use two very different preventative strategies: doxyPEP and one-time vaccination to reduce infection:

- From the theory of planned behavior, intent is the single most important driver of using a given preventative measure. Why would half of respondents have intent to use doxyPEP, but only a small proportion have ever used it? The analysis could focus on the differences in determinants of those who have intent versus actual use, given the limitation that doxyPEP is not (yet) recommended in the UK and the questions were not asked on the same prevention strategy (i.e., “any antibiotic” for use versus “doxycycline” specifically for intent).

- Alternatively, GBMSM are confronted with two prevention strategies: doxyPEP or vaccination. What are the determinants of intending to use only doxyPEP, only vaccination or both versus none? This multinomial outcome could be more helpful in planning provisions or tailoring needs.

I would suggest that the authors focus on one of these research questions above.

It also seems that transgender women were included in the population (ln 109), but this could be a mistake. This key population is completely different than GBMSM and should either be considered separately (i.e., stratified) or not included in analysis.

Minor comments:

- ln 32. “Emerging evidence”? There were several RCT that confirmed this. Not sure if this evidence is still emerging.

- ln 49. What were the numbers and % of individuals responding with intention?

- ln 53. How are the authors defining “high levels”?

- ln 55-57. This sentence is ambiguous and cannot be concluded from the data provided. Please omit.

- ln 59. “unprecedented”? Were such increases not observed in the 1970s?

- ln 67. Should be “selection of AMR bacteria.”

- ln 69-74. Although important, it is unclear how these sentences relate to the research objective. Consider removing.

- ln 90. “… in preparation for potential use…” What does this mean?

- General comments in the methods: It would be helpful to state which questions were asked and how your definitions were made. Then have a separate section for the statistical analysis.

- Statistical analysis. It is unclear how the multivariable model was constructed. It seems that there was an a priori selection of covariables. Then the authors added all covariables with a p-value <0.05 in univariable analysis? Was there further selection (i.e., backwards stepwise selection) to create the final model? And how were issues such as collinearity or model fit assessed? (I am guessing that collinearity was the reason why the association between markers of sexual risk and doxyPEP ITU in the multivariable model, ln 214-215). From the information provided, I cannot replicate these analyses.

- ln 155. I would consider using a flow diagram of participant selection for the main manuscript.

- ln 162. I would plainly state, “the period from August 2023 to questionnaire” as “lookback period” is somewhat ambiguous.

- ln 172. Curious if there were any questions on how these individuals procured doxycycline (formally or informally)?

- Tables 2-4. It gives the covariables on which models were adjusted, but are these correct? Are these in addition to all other covariables listed in the column for the multivariable model?

- ln 208. What is the “age-adjusted model”?

- ln 240. Was “recommended for use in the UK” a part of the question in the questionnaire? If not, please omit.

- ln 250. Is there empirical evidence that doxyPEP could drive AMR?

- ln 259. “… but this may reflect the wider availability of STI testing in the UK”. Not sure I follow this argument.

- ln 287. How can we inform on the risks when we do not completely know what they are? This uncertainty might need to be highlighted.

- ln 305-326. HIV-PrEP is a very different in many respects to doxyPEP and the comparison is difficult to make. Moreover, doxyPEP is part of the CDC guidance in the United States (not just in “some US cities.”, ln 323). It is unclear how these data will be used to plan provisions (and this can be the focus point of an entirely new manuscript). Many of the policy recommendations are already stated in the conclusion paragraph. The paragraph needs to be removed.

Reviewer #3: Thank you to the authors for conducting this work. These estimates are highly informative for both research and policy development on novel biomedical interventions for bacterial STI preventions. I have a few comments regarding some of the language used in the article as well the clarifying some of the variables and modelling strategy.

1. These results are presented as among “gay, bisexual, and other men who have sex with men” but the inclusion criteria states that transgender women and other gender diverse individuals were included in the study. Is it possible to include further details on recruitment in the methods section (e.g., was the survey advertised as for gbMSM and some trans women ended up participating?) and clarify the language used in the article? In addition, please clarify how gender was measured (e.g., two step method vs single question).

2. Similarly, I understand the need to dichotomize the sexual orientation variable for the analyses, but it would be helpful to specify in the methods that the “bisexual” category is really an “other” category, and it may be worth considering re-labelling it.

3. Can you include more details in the methods section on how variables were selected for the multivariable modelling? Why were only sociodemographic variables that were statistically significant at p<0.05 used in the adjusted model? It appears as though only variables significant at p<0.05 in the bivariable models were then moved to an adjusted estimate, is that correct?

4. Please mention in the methods that the composite of sexual risk behaviour was a binary indicator (rather than a continuous or ordinal one).

5. Can you clarify the modelling strategy – is this one multivariable model per outcome or one adjusted model per predictor? It appears as though these are separate, adjusted models for each exposure-outcome association, but this would be helpful to clarify in the methods.

6. PLOS authors have the option to publish the peer review history of their article (what does this mean?). If published, this will include your full peer review and any attached files.

**Do you want your identity to be public for this peer review?** For information about this choice, including consent withdrawal, please see our Privacy Policy.

Reviewer #1: No

Reviewer #2: No

Reviewer #3: No

---

## [Decision Letter · Decision Letter 1]

26 Sep 2024

Knowledge, uptake and intention to use antibiotic post-exposure prophylaxis and meningococcal B vaccine (4CMenB) for gonorrhoea among a large, online community sample of men and gender-diverse individuals who have sex with men in the UK

PGPH-D-24-01329R1

Dear Ms Ogaz,

We are pleased to inform you that your manuscript 'Knowledge, uptake and intention to use antibiotic post-exposure prophylaxis and meningococcal B vaccine (4CMenB) for gonorrhoea among a large, online community sample of men and gender-diverse individuals who have sex with men in the UK' has been provisionally accepted for publication in PLOS Global Public Health.

Best regards,

Tsz Ho Kwan, PhD

Academic Editor

The reviewer comments have been adequately addressed.

Reviewer Comments (if any, and for reference):

Reviewer's Responses to Questions

**Comments to the Author**

1. If the authors have adequately addressed your comments raised in a previous round of review and you feel that this manuscript is now acceptable for publication, you may indicate that here to bypass the “Comments to the Author” section, enter your conflict of interest statement in the “Confidential to Editor” section, and submit your "Accept" recommendation.

Reviewer #1: All comments have been addressed

Reviewer #3: All comments have been addressed

2. Does this manuscript meet PLOS Global Public Health’s publication criteria? Is the manuscript technically sound, and do the data support the conclusions? The manuscript must describe methodologically and ethically rigorous research with conclusions that are appropriately drawn based on the data presented.

Reviewer #1: Yes

Reviewer #3: Yes

3. Has the statistical analysis been performed appropriately and rigorously?

Reviewer #1: Yes

Reviewer #3: Yes

4. Have the authors made all data underlying the findings in their manuscript fully available (please refer to the Data Availability Statement at the start of the manuscript PDF file)?

Reviewer #1: Yes

Reviewer #3: Yes

5. Is the manuscript presented in an intelligible fashion and written in standard English?

Reviewer #1: (No Response)

Reviewer #3: Yes

6. Review Comments to the Author

Reviewer #1: Please proofread as the edited text contains a few minor typos (eg, line 122 "sex a birth")

Reviewer #3: (No Response)

7. PLOS authors have the option to publish the peer review history of their article (what does this mean?). If published, this will include your full peer review and any attached files.

**Do you want your identity to be public for this peer review?** For information about this choice, including consent withdrawal, please see our Privacy Policy.

Reviewer #1: No

Reviewer #3: No
